

# Effectiveness of isolation policies in schools: evidence from a mathematical model of influenza and COVID-19

Adam A.C. Burns[1] and Alexander Gutfraind[1,2]

[1] Division of Hepatology, Department of Medicine, Loyola University of Chicago, Maywood, IL, USA
[2] Division of Epidemiology and Biostatistics, School of Public Health, University of Illinois at Chicago, Chicago, IL, USA

Corresponding author
Alexander Gutfraind,
agutfrai@uic.edu

## ABSTRACT

**Background:** Non-pharmaceutical interventions such as social distancing, school closures and travel restrictions are often implemented to control outbreaks of infectious diseases. For influenza in schools, the Center of Disease Control (CDC) recommends that febrile students remain isolated at home until they have been fever-free for at least one day and a related policy is recommended for SARS-CoV-2 (COVID-19). Other authors proposed using a school week of four or fewer days of in-person instruction for all students to reduce transmission. However, there is limited evidence supporting the effectiveness of these interventions.

**Methods:** We introduced a mathematical model of school outbreaks that considers both intervention methods. Our model accounts for the school structure and schedule, as well as the time-progression of fever symptoms and viral shedding. The model was validated on outbreaks of seasonal and pandemic influenza and COVID-19 in schools. It was then used to estimate the outbreak curves and the proportion of the population infected (attack rate) under the proposed interventions.

**Results:** For influenza, the CDC-recommended one day of post-fever isolation can reduce the attack rate by a median (interquartile range) of 29 (13–59)%. With 2 days of post-fever isolation the attack rate could be reduced by 70 (55–85)%. Alternatively, shortening the school week to 4 and 3 days reduces the attack rate by 73 (64–88)% and 93 (91–97)%, respectively. For COVID-19, application of post-fever isolation policy was found to be less effective and reduced the attack rate by 10 (5–17)% for a 2-day isolation policy and by 14 (5–26)% for 14 days. A 4-day school week would reduce the median attack rate in a COVID-19 outbreak by 57 (52–64)%, while a 3-day school week would reduce it by 81 (79–83)%. In both infections, shortening the school week significantly reduced the duration of outbreaks.

**Conclusions:** Shortening the school week could be an important tool for controlling influenza and COVID-19 in schools and similar settings. Additionally, the CDC-recommended post-fever isolation policy for influenza could be enhanced by requiring two days of isolation instead of one.

## INTRODUCTION

Respiratory infections are the leading cause of death in low- or middle-income countries and account for an estimated four million deaths annually (*Mathers & World Health Organization, 2008*). For rapidly emerging outbreaks such as novel strains of influenza or SARS-CoV-2, pharmaceutical measures may be unavailable or ineffective, in which case non-pharmaceutical intervention measures (NPIs) may be the first line of response against infection (*Ferguson et al., 2006*; *Cowling et al., 2020*). However, there is currently an acute need to develop and validate NPIs for COVID-19 (the disease caused by the SARS-CoV-2 virus) (*Brauner et al., 2020*) and refine the understanding of NPIs for controlling influenza (*WHO, 2019*).

Our study directly addresses this need by using a computational model to examine the impact of several NPIs, focusing on a shortened school week and symptom-based isolation policies. The shortened school week policy involves closure of the school for additional days to extend the weekend (e.g., closure every Thursday and Friday) (*Faherty et al., 2019*; *Karin et al., 2020*), thus creating a period of additional physical isolation between the students, and possibly conducting at-home learning on those days. The symptom-based isolation policy involves isolating individuals at the onset and for the duration of fever symptoms, normally followed by additional days of isolation. Additionally we considered increasing compliance and monitoring, reducing cross-grade contact, and complete weekend quarantine policies.

For controlling pandemic and seasonal influenza outbreaks, the symptom-based NPI with 1 day of post-fever isolation is currently recommended by the US Centers of Disease Control and Prevention (CDC) (*Centers for Disease Control & Prevention, 2018*; *Qualls et al., 2017*), and is referred to as fever absenteeism or a return-to-school policy (*Miller et al., 2013*). The buffer period of 1 day reduces transmission from infectious students in situations where their fever symptoms temporarily subside or when they continue shedding the virus at the end of the course of illness (*Carrat et al., 2008*).

For COVID-19 outbreaks, there are currently divergent recommendations from public health authorities, and there have been several changes in guidelines. For symptomatic patients, the WHO currently recommends 10 days of isolation after symptom onset, plus at least 3 additional days without symptoms, while for asymptomatic patients it recommends 10 days after positive test for SARS-CoV-2 (*WHO, 2020*). The CDC recommends that patients with COVID-19 symptoms isolate until they have met all three conditions: (1) they have been fever-free for 24 h without the use of fever-reducing medications, (2) it has been 10 days since the onset of their symptoms and (3) their COVID-19 symptoms have been improving (*Centers for Disease Control & Prevention, 2020*). Finally, the Swedish Public Health Agency (SPHA), which previously required 2 days free of symptoms, currently gives the schools control in establishing their own isolation policies ("Information till förskola, grundskola och gymnasier om covid-19—Folkhälsomyndigheten", http://www.folkhalsomyndigheten.se/smittskydd-beredskap/utbrott/aktuella-utbrott/covid-19/verksamheter/information-till-skola-och-forskola-om-den-nya-sjukdomen-covid-19/).

To evaluate the various NPIs, our model computationally simulates outbreaks of influenza and COVID-19 in school settings, and then looks at the effect of isolation policies on the attack rate (i.e., the proportion of the student population infected over the duration of the outbreak) and the outbreak curve (i.e., prevalence of infected students at each day of the outbreak). Our work builds upon previous studies that have applied mathematical models to school-based influenza transmission (*Halloran et al., 2008*; *Coburn, Wagner & Blower, 2009*; *Chao et al., 2010*; *Christensen et al., 2010*). The policy of symptom monitoring, which isolates contacts after onset of symptoms, has been examined computationally and shown to be sufficient for controlling certain outbreaks (*Peak et al., 2017*). Several studies also modeled non-pharmaceutical interventions such as closures but not absenteeism policies (*Andreasen & Frommelt, 2005*; *Milne et al., 2008*; *Cauchemez et al., 2009*; *Sasaki et al., 2009*; *Rhodes & Hollingsworth, 2009*; *Bansal et al., 2010*; *Lee et al., 2010*; *Christensen et al., 2010*; *Stebbins et al., 2011*; *Araz et al., 2012*; *Jackson et al., 2013*; *Diedrichs, Isihara & Buursma, 2014*; *Eames, 2014*). We retrieved all studies that evaluated isolation policies by using a broad search on PubMed and found that many modeling studies assumed isolation for a fixed interval following diagnosis but not in a symptom-dependent way (*Halloran et al., 2008*). We also found several comprehensive computational studies of school closures and isolation of infected students (*Halloran et al., 2008*; *Kelso, Milne & Kelly, 2009*), but did not find evaluation of symptom-based isolation policies. The policy of shortening school weeks was recently examined in a modeling study (*Karin et al., 2020*) that found that closures for 10 days followed by 4 days of schooling could be effective in controlling COVID-19, but our model is the first to examine the modification of a standard 5-day school schedule.

Despite the promise of post-fever isolation for both COVID-19 and influenza, we hypothesized at the outset of this project that the policy would merely have a small effect in controlling outbreaks. We speculated that the CDC-recommended single day of post-fever isolation might not be enough to achieve a meaningful reduction in influenza transmission. Furthermore, parental non-compliance may make the policy ineffective. As compared to influenza, the policy would be less effective in COVID-19 outbreaks since children have a higher rate of asymptomatic infections (*Poletti et al., 2020*). On the other hand, if the policy were proven effective for either outbreak, it is not known whether 1 day of post-fever isolation is optimal, and the model could help determine if additional days of isolation would be beneficial. Additionally, we hypothesized that the short school-week policy, although disruptive, may be more effective for both infections since it could be easier to enforce than symptom-based isolation. In the case of COVID-19, the policy has the advantage of not being affected by the low rate of symptomatic infections among children (*Poletti et al., 2020*).

## MATERIALS AND METHODS

We use a deterministic compartmental dynamical model known as the Susceptible, Exposed, Infectious, Recovered (SEIR) model that tracks the number of individuals of various cohorts immunological states, and degree of isolation for each day during an outbreak (Fig. 1A; Table 1). This class of SEIR models has been used extensively to model

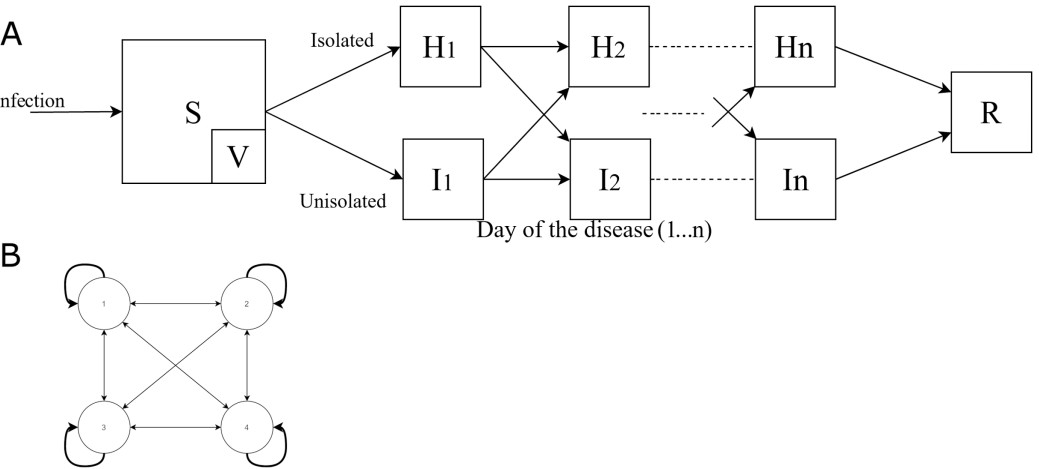

**Figure 1 The model.** (A) Dynamics of the outbreak model and its major variables. S, susceptible; H, infected isolated; I, infected unisolated; R, recovered; and V, vaccinated or immune (if a vaccine is available). The students are stratified to cohorts, and n daily disease stages distinguished by severity of symptoms and viral shedding (9 daily stages of viral shedding for influenza and 32 stages for COVID-19, see Supplemental Digital Content—Expanded Methods). The cohort structure and other parameters are adjustable to model outbreaks in different settings and by different pathogens. Vaccination is available only for some outbreaks and generally has incomplete coverage and efficacy. (B) Illustration of the typical connectivity pattern between cohorts in the school where the thickness of the arrows illustrates the greater risk of transmission within cohort than across cohorts (shown is the case of four cohorts).

**Table 1 Variables of the model. Index $i$ indicates the cohort (normally student grade) and $d$ the day of infection (running from 1 to 9, or 32 for influenza and COVID-19, respectively).**

| Variable | Interpretation |
| --- | --- |
| $S_i$ | Naïve persons in cohort $i$ (for immunity, $\tilde{S}_i(0) = S_i(0)(1 - v v_e)$, where $v$ is the fraction vaccinated and $v_e$ is the vaccine efficacy |
| $I_{i,d}$ | Infected persons in cohort $i$ on day $d$ of infection who are not isolated |
| $H_{i,d}$ | Similarly, but who are *isolated* (e.g., at home) |
| $R_i$ | Recovered in cohort $i$ |

influenza (*Andreasen & Frommelt, 2005*; *Coburn, Wagner & Blower, 2009*; *Modchang et al., 2012*; *Chen et al., 2019*) and COVID-19 (*Wang et al., 2020*; *He, Peng & Sun, 2020*), and we extended the SEIR framework in order to account for isolation policies (Box 1).

Our model is particularly novel as it further stratifies the population by both the day of their infection, their location (isolated at home vs. not isolated in school) and their grade, with students in the same grade generally having closer contacts to peers in the same grade (*Glass & Glass, 2008*) (see Fig. 1B, Article S1—Expanded Methods for details). In the model, the day of infection determines the rate of virus shedding and the probability of symptoms, which then influences the likelihood of either isolating at home or returning to school. The probability of isolating was also based on the stage of the illness, as well as the isolation policy. The model allowed a vaccine to be received by some students, if

> **Box 1 Our modeling framework—the multi-state discrete time SEIR infection model stratified by cohort and day of infection.**
>
> Cohorts normally correspond to student grades but they could also include part of each grade and school staff. Variables are defined in Table 1 and are dependent on time $t$. Index $i$ runs over all cohorts in the school and index $d \in \{1 \ldots d_f\}$ over the days from day 1 after infection to recovery. Return rates from isolation at day $r_d(p)$ depend on symptom isolation policy $p$, $\delta_{1,d}$ is the Kronecker delta function (=1 if $d$ is 1, and otherwise zero). $c_{i,j}$ is the connectivity between cohorts $i$ and $j$ and $s_d$ is the viral shedding at day $d$ of infection. Estimates for the parameters are in Appendix Tables S2, S3 and S4.
>
> $\Delta S_i = -bS_i\lambda_i$, where $\lambda_i = \Sigma_{j,d}c_{i,j}s_dI_{j,d}$ is the force of infection
>
> $H_{i,d} = (1 - r_d(p)) (H_{i,d-1} + I_{i,d-1}) (1 - \delta_{1,d})$
>
> $I_{i,d} = r_d(p) (H_{i,d-1} + I_{i,d-1}) (1 - \delta_{1,d}) + \delta_{1,d}bS_i\lambda_i$
>
> $\Delta R_i = H_{i,d_f} + I_{i,d_f}$

the vaccine is available ahead of the outbreak, attaining partial protection against the infection. The model also considered any pre-existing immunity, the rate of symptomatic infections, and school holidays. Policy analysis used the setting of a typical school (6 grades with 70 students each) (*Common Core of Data (CCD)*), but also considered alternative scenarios with larger schools (140 students per grade).

The model was validated on outbreaks of influenza and COVID-19 in schools and shown to match the peak and duration of the outbreak curves, and the overall attack rates of the student population. The validation data was from two outbreaks of pandemic influenza (*Smith et al., 2009*; *Bhattarai et al., 2011*), one outbreak of seasonal influenza (*McCann et al., 2014*), and one outbreak of COVID-19 ("3 teachers in shuttered Jerusalem school have virus; pupils, staff to be tested", https://www.timesofisrael.com/3-teachers-in-shuttered-jerusalem-school-have-virus-pupils-staff-to-be-tested/). Model parameter ranges were derived from published sources and by calibration to data using a stochastic optimization algorithm.

The estimated parameters were: (a) the overall transmissibility of the virus, (b) the relative transmissibility on weekends and (c) during school closures, (d) the effect of the season on transmission, (e) parental attention to symptoms, and (f) compliance to the symptom isolation policy (see Appendix, Table S2). The transmissibility (a) was calibrated separately for influenza and COVID-19.

To ensure that the results were robust to uncertainty in parameter values, we then simulated the epidemic 500 times per scenario to account for possible difference between schools and seasons, with normally distributed values for parameters such as the start day in the year, contact rate between cohorts and others, and reported the median and the interquartile ranges. All modeling and statistical analysis used the RStudio Integrated Development for R. RStudio, Inc., Boston, MA. Further details on the model, including the equations and the parameter values, are provided in Article S1—Expanded Methods. In addition to this text, we provide the source code for the model, the associated data, and the expanded results (https://github.com/sashagutfraind/feverfighter). A live configurable dashboard for applying the model is also available (https://epi1.shinyapps.io/FeverFighter/).

**Table 2 Relative effectiveness of isolation policies. Median effect (interquartile range).**

| Outbreak | Policy option | Attack rate | | Outbreak duration | |
|---|---|---|---|---|---|
| | | Baseline (%) | % decrease | Baseline (days) | % decrease |
| *Flu* | One day isolation (CDC guideline) | 25 | 29 (13–59)% | 82 | 1 (–2 to 16)% |
| | Two day post-fever isolation | | 70 (55–85)% | | 18 (6–66)% |
| | Four day school week | | 73 (64–88)% | | 20 (11–55)% |
| | Three day school week | | 93 (91–97)% | | 99 (82–100)% |
| *COVID-19* | One day isolation | 11.3 | 7 (5–14)% | 138 | 1 (1–4)% |
| | Two day post-fever isolation | | 10 (5–17)% | | 1 (1–4)% |
| | 14 day post-fever isolation | | 14 (5–26)% | | 4 (3–7)% |
| | Four day school week | | 57 (52–64)% | | 22 (12–26)% |
| | Three day school week | | 81 (79–83)% | | 46 (33–52)% |

Using the model we considered the effect of two key control policies, fever-based isolation and a shortened school week. For fever-based isolation we evaluated the effect of stricter compliance, which could be attained by remote monitoring, help in maintaining home isolation or penalties for non-compliance. We also considered the effect of increasing the monitoring of symptoms, which could be attained through training of the parents and distribution of free thermometers. We also considered supplemental policies: subdividing students into cohorts of half the normal size, reducing contacts between cohorts, and enforcing strict quarantines on weekends. We evaluated all policies using three outcome measures: (1) the attack rate—the proportion of the population infected during the outbreak, (2) the outbreak duration—the number of days with more than one infected student, and also (3) the peak number of simultaneously infected—a measure of the burden on the caregivers and the healthcare system.

## RESULTS

Our analysis considered the effect of symptom-based isolation and alternative school schedule policies on influenza and SARS-CoV-2 infections. The effect of these policies is summarized in Table 2 and described in detail below.

### Influenza

[1] In the simulations, the input parameters were chosen at random but symmetrically around the calibrated values. However, the effect of deviations from the calibrated values is asymmetrical on the outbreak, causing the median value of measures like the attack rate in the 500 simulations to differ from the baseline value (25% attack rate for influenza and 11.3% attack rate for SARS-CoV-2). Reducing the magnitudes of the deviations removes this effect.

We calibrated our model's parameters to outbreaks of pandemic and seasonal influenza and then calculated each isolation policy 500 times.[1] The different runs varied parameters that normally vary from year to year (e.g., outbreak start day and compliance with policy). In the baseline scenario of pandemic influenza and no isolation policy, our model has a median attack rate of 24.5 (Interquartile Range, IQR: 16.6–28.1)%. Implementing a one- and two-day isolation policy following fever, our model predicts a decrease in the attack rate to 17.2 (9.9–21.4)% and 7.4 (3.7–11.1)%, respectively (Fig. 2).

Furthermore, the model predicts that a 2-day isolation policy reduces the peak prevalence (i.e., simultaneously infected students) from 20 (13–25) to 5 (2–8) and the outbreak duration from 82 (78–84) to 67 (28–77) days. Our model suggests that the 2-day
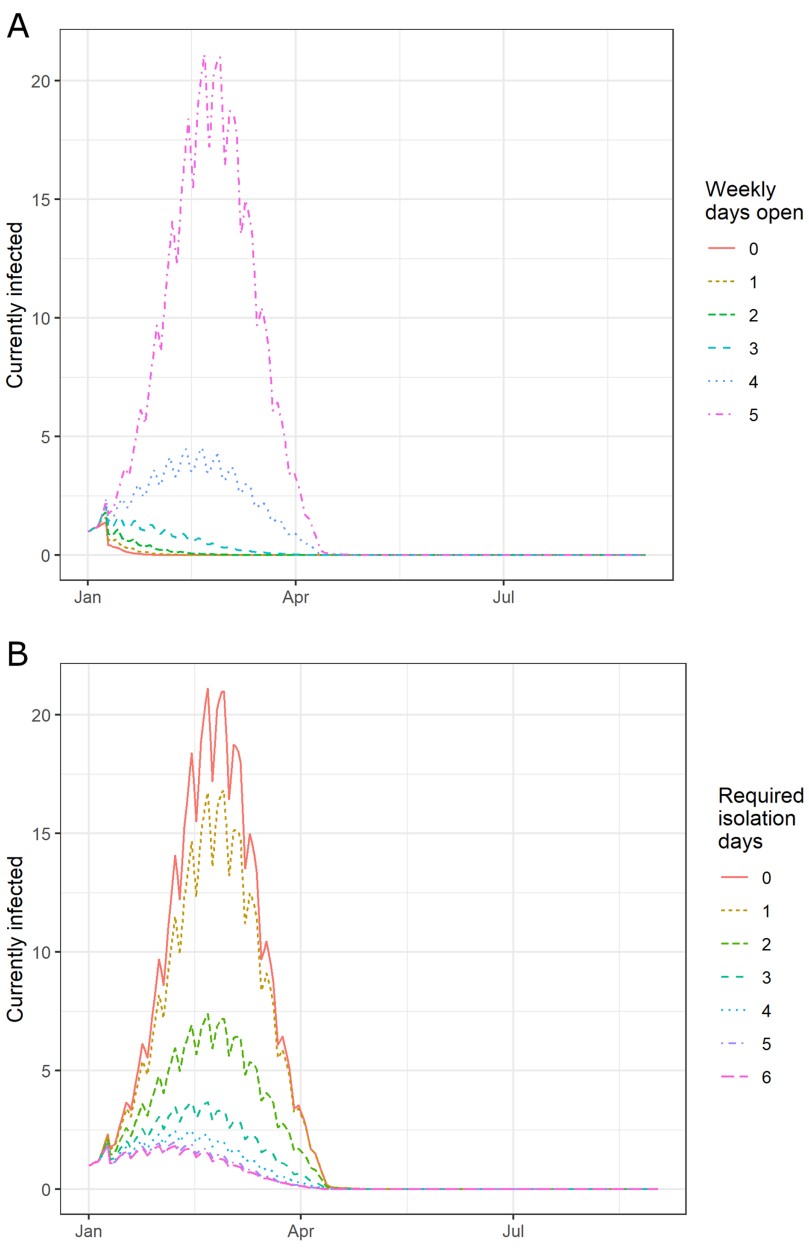

**Figure 2 The effect of requiring isolation after the last fever event in a median US school experiencing an outbreak of influenza.** (A) Fever isolation and (B) shortened in-person school week. Vertical axis indicates daily prevalence and ripples are due to weekends and closures. Summer holiday starts June 17 and reduces transmission. Increasing the required days of isolation or shortening the in-person school week reduces the peak infected and the number concurrently infected. Only shortening the in-person school week reduces the duration of the outbreak.

[2] The policy appears to be less effective in the presence of immunity due to a paradoxical effect noted in Ref (*Echevarria et al., 2015*). In both the no immunity and immunity scenarios we assume an attack rate of 25%, but in the presence of immunity, the virus needs to have higher transmissibility to achieve this attack rate and is thus harder to control.

policy is still effective even when the student population has pre-existing immunity or has been vaccinated (mean vaccination rate of 80% and efficacy of 50%), reducing the attack rate to 13 (4.2–23.8)%[2].

We have also examined the policy of reducing the number of days of in-person study. A policy of a 4-day school week gives a 73% reduction in the attack rate from the baseline, to 6.8 (3.3–8.8)%, and a 3-day school week gives a 93% reduction to 1.8 (0.9–2.3) (Fig. 2B). The 2-day of post-fever isolation and 3-day week policies could be combined additively with 2 days of post-fever isolation to give attack rates of 2.1 (1.0–3.3)% and 0.9 (0.5–1.2)%, respectively.

### COVID-19 (SARS-CoV-2)

In our COVID-19 outbreak scenarios, in the absence of policy measures, the baseline attack rate was 10.0 (8.7–11.3)% in 500 simulations. As expected based on the lower rate of symptomatic infections in COVID-19 as compared to influenza, post-fever isolation policy is less effective for COVID-19: 1, 2, and 14 days of isolation yielded attacks rates of 9.4 (8.3–10.6)%, 9.2 (8.0–10.6)%, and 8.5 (7.4–9.7)%, respectively (Fig. 3A). Additionally, our model had a baseline outbreak duration of 138 (135–140) days and yielded outbreak durations of 137 (133–139), 136 (132–139), and 132 (128–134) days for 1, 2, and 14 days of post-fever isolation. Thus when compared to the influenza outbreaks, this type of policy was less effective at attack rate or outbreak duration inhibition.

Evaluating the policy of shortening the school week, our model found that using 4 and 3 days of in-person schooling yields attack rates of 4.4 (3.7–4.9)% and 2.0 (1.7–2.2)%, respectively (Fig. 3B). When the student population is presumed to be immune (*Sette & Crotty, 2020*) or vaccinated at a rate of 80% and with an efficacy of 70% (Standard Deviation, SD: 20%), the model predicts that the policy is still effective: for the 4- and 3-day school week, the model predicts a median attack rate of 5.1 (1.8–15.2)% and 2.7 (1–6.6)%, respectively.

For both influenza and COVID-19, we examined additional policy options to complement the two main policies of post-fever isolation and shortening the school week. In the case of symptom-based isolation for influenza, measures that increase compliance and symptom monitoring were found to be effective: a 25% increase in attention and compliance reduces the attack rate to 3.4 (1.5–5.2)% and 7.3 (3.5–11.2)%, respectively. For both infections, reducing contacts between student grades within a school was found to be very effective and could be complementary to the main policies. Implementing strict quarantines on weekends were also found to be effective.

## DISCUSSION

Outbreaks of acute respiratory infections such as influenza and the novel COVID-19 require an expansion of the available infection control policies. Here we report evidence in support of several such policies across outbreak scenarios and settings. For influenza, requiring isolation for fever is expected to reduce the typical attack rate by 29 (13–59)% and 70 (55–85)% with 1 and 2 days of post-fever isolation, respectively. This indicates that the CDC-recommended policy for schools, based on a single day of post-fever isolation, could be enhanced by requiring a second day of isolation. The result also holds in seasonal influenza in which vaccination is implemented. The isolation policy could be further strengthened by reducing contact between students during weekends. Using a

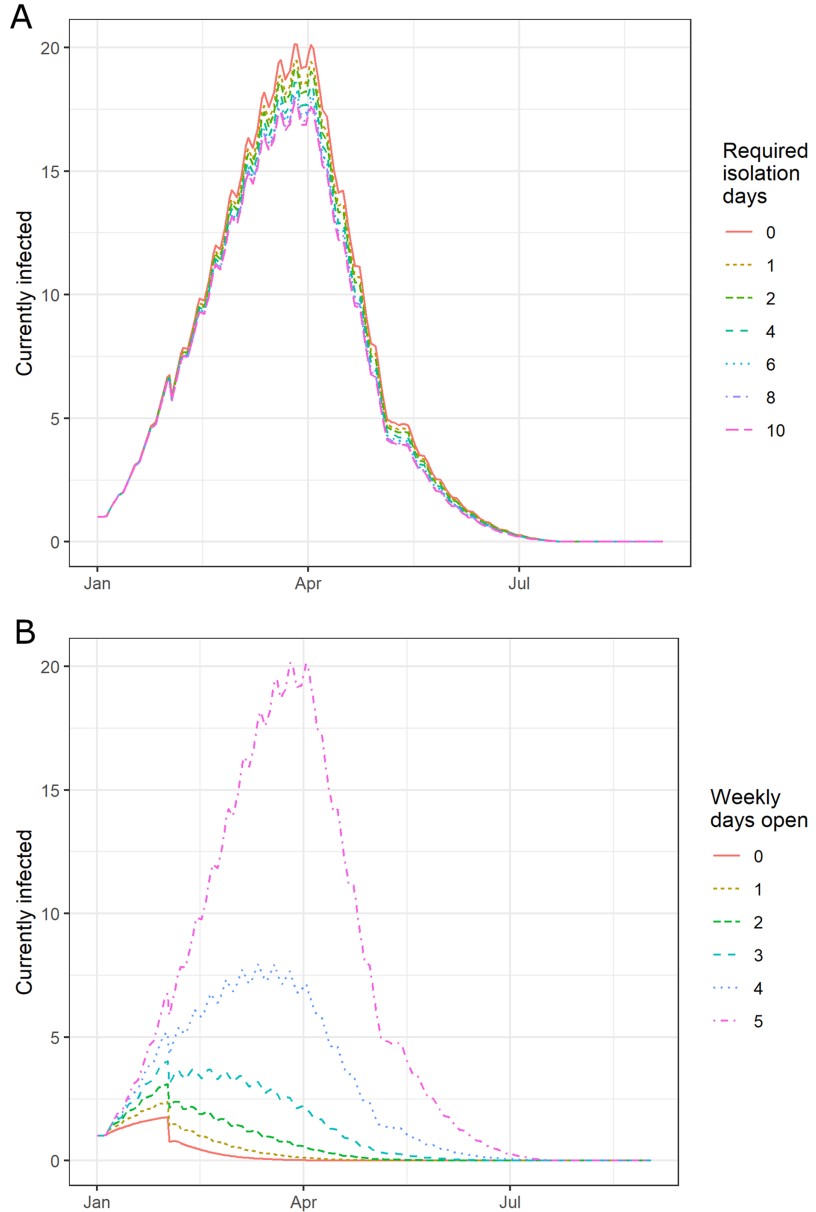

**Figure 3 The effect of requiring isolation after the last fever event in a median US school experiencing an outbreak of COVID-19.** (A) Post-fever isolation and (B) in-person school-week reduction policies on a median US school experiencing an outbreak of COVID-19. Vertical axis indicates daily prevalence as in Fig. 2. Increasing the number of post-fever isolation days has little effect on the outbreak. Reducing the number of school days that students physically go to school each week reduces the peak number of infected, the number concurrently infected, and the duration of the outbreak.

shorter in-person school week (i.e., through longer weekends or remote learning) would also reduce the attack rate by 73 (64–88)% and 93 (91–97)% with 4 and 3-day school weeks, respectively. The high percentage of reduction arises because these measures are expected to bring the epidemic under the outbreak threshold, and assumes no re-introduction of the infection from outside the school.

While we expected symptom-based isolation to be ineffective for COVID-19 since children are commonly asymptomatic (*He et al., 2020*), it can still help in reducing outbreaks. Many authorities, including the CDC and WHO, recommend that those with symptoms isolate for a minimum of 10 days after symptom onset (*WHO, 2020*) (*Centers for Disease Control & Prevention, 2020*). Indeed, we found that a 1-day post-fever isolation policy would reduce the attack rate in schools by 7 (5–14)%, and with 14 days of fever isolation we estimated that the attack rate would change by 14 (5–26)%. The result shows that symptom-based isolation cannot be relied upon as a central policy in outbreak control for COVID-19, but it is not futile. Current CDC and WHO isolation policies for COVID-19 are expected to have effectiveness between the 1-day and 14-day fever-based isolation policies above, but require only 1 day of isolation following any fever (CDC) and 3 days of isolation following cessation of symptoms (*WHO, 2020*). We found that shortening the school week does reduce the total number infected in an outbreak by 57 (52–64)% and 81 (79–83)%, with four and 3 days of in-person schooling, respectively.

There are several possibilities for how shortening of the school week may reduce the transmission. It could: (1) reduce the amount of time the infected and uninfected populations mix, or (2) create an extended period of isolation of the infectees. To identify the cause, we held (1) constant by using a policy of 4 days of schooling per week but varied (2). Namely, we performed simulations where an outbreak was controlled using three different regimens with a shortened school week: (a) a regimen where over a 2 week period, the school is closed on days 4, 6, 8, 10, 12, and 14, (b) closing for 3 days every Friday (i.e., closures on days 5, 6, 7, 12, 13, and 14), and a school that opens for 8 days and closes for 6 days straight. The attack rate was lowest in the third policy: for influenza, the regimens produced attack rates of 6.8%, 5.3% and 3.8%, and for SARS-CoV-2: 4.6%, 4.7%, and 3.7%. We conclude that the short school week acts both to reduce the amount of mixing and separately by creating a continuous period of isolation.

Generally speaking, policies that isolate the infected, of which symptom-based isolation is a sub-type, are more preferable to closures. Unlike closures, symptom-based isolation allows healthy people in the community to continue living their lives uninhibited and reduces the considerable societal cost of school closures (*Viner et al., 2020*). Consequently, a standing policy of isolation of infected individuals could potentially be sustained indefinitely. Shorter school weeks do affect all students, but they are less disruptive than full closures. They can also be maintained for extended times during the peak of the outbreak season, particularly if school days are replaced with remote learning (self study or e-learning). In severe outbreaks, a combination of policies would provide the best outcomes. High compliance by families would be critical to ensuring that the students are strongly isolated when the school is closed, and therefore families need to be supported.

While our model is driven by virological data and calibrated to several outbreaks, a few limitations are inherent in our approach. First, the effect of any policy depends on the context where it would be applied. The details of the school or institution would matter, and therefore, we provide an online version of the model, which can be calibrated for multiple situations. Better calibration data would help improve the calibration of

the model, and in certain datasets it was not established if the absence was due to the outbreak or due to other causes. It may be possible to apply our findings to school-like contexts such as workplaces, prisons, or even the broader community, but such settings have significant features that may confound our findings. Lastly, symptom parameter information is based on average values for the population and it is expected that inter-individual and demographic variability might have some effect on outcomes. Future studies should attempt to evaluate isolation policies with agent-based models (e.g., (*Chao et al., 2010*; *Gutfraind et al., 2015*)) that can capture inter-individual variability in health trajectories and the network structure of the population(*Lloyd-Smith et al., 2005*; *Volz et al., 2011*). Despite these limitations, our model captures essential aspects of acute respiratory outbreaks including progression through stages, the population structure and symptom trajectories.

## CONCLUSION

In this study we have created a model of transmission of respiratory infection and considered the effects of two isolation policies. We confirmed that symptom-based policies would be effective in controlling influenza in a variety of scenarios. Furthermore for influenza outbreaks, we recommend that isolation is maintained for at least 2 days following the last day of fever. For both influenza and COVID-19 we found that using a shortened school-week of 4 days instead of 5 days could be effective in reducing the attack rate, and additional days would increase the effect. Policymakers tackling the influenza and COVID-19 outbreaks should consider implementing these policies for controlling outbreaks in schools and other settings.

### Funding

This research is supported by the US National Institutes of Health (NIH) grant R01GM121600. The funders had no role in study design, data collection and analysis, decision to publish, or preparation of the manuscript.

### Grant Disclosures

The following grant information was disclosed by the authors:
US National Institutes of Health: R01GM121600.

### Competing Interests

The authors declare that they have no competing interests.

### Author Contributions

- Adam A.C. Burns conceived and designed the experiments, performed the experiments, analyzed the data, prepared figures and/or tables, authored or reviewed drafts of the paper, and approved the final draft.

- Alexander Gutfraind conceived and designed the experiments, performed the experiments, analyzed the data, prepared figures and/or tables, authored or reviewed drafts of the paper, and approved the final draft.

## Data Availability

Detailed methods are available as a Supplemental File.

Code and additional data is available at GitHub: https://github.com/sashagutfraind/feverfighter.

A live configurable dashboard for applying the model is available at shinyapps.io: https://epi1.shinyapps.io/FeverFighter.

## Supplemental Information

Supplemental information for this article can be found online at http://dx.doi.org/10.7717/peerj.11211#supplemental-information.

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
