# Peer review of "Effectiveness of isolation policies in schools: evidence from a mathematical model of influenza and COVID-19"

_PeerJ, doi:10.7717/peerj.11211_

## Round 0.1 · original submission · Major Revisions

After reading your work and the reviewers' comments, I think your manuscript has scientific merit to be published in PeerJ, once you assessed all their questions.

Reviewer 1 ·

Basic reporting

The authors used mathematical models to evaluate the effects of different school-based control measures on epidemics of influenza and COVID-19. They found that isolating symptomatic students could reduce attack rate of influenza but not COVID-19, which makes sense because of the common pre-symptomatic transmission of COVID-19. The authors also found that shortening school open days could substantially reduce the attack rate of both influenza and COVID-19, which is interesting and perhaps need more elaboration of why.

Experimental design

Overall I think the design of study was reasonable and valid. I would suggest to define the outcome measures first in the methods section and provide more details/analysis to elaborate how shortening school open days could reduce attack rate (e.g. the mechanisms).

Validity of the findings

Overall I think findings from the study were reasonable and valid. The findings that shortening school open days can reduce attach rate for influenza and COVID-19 is particularly interesting, but the machinist behind this is bit unclear and may need to be validated by intermediate results from the model simulations. It would be helpful if the authors could see which steps result in the reduction in attack rate, e.g., the exposure time of all students or just the infected students. These may have more implications on targeted controls - which would be more effective, control exposure or control sources?

Additional comments

1. I would suggest to define the outcome measures first in the methods section and provide more details/analysis to elaborate how shortening school open days could reduce attack rate (e.g. the mechanisms).
2. The findings that shortening school open days can reduce attach rate for influenza and COVID-19 is particularly interesting, but the machinist behind this is bit unclear and may need to be validated by intermediate results from the model simulations. It would be helpful if the authors could see which steps result in the reduction in attack rate, e.g., the exposure time of all students or just the infected students. These may have more implications on targeted controls - which would be more effective, control exposure or control sources?
3.I don't think that I agree with the authors that NPIs lack of sufficient evidences of effectiveness. It’s not clear of the direct and indirect effects of these associations.
4. Did the author look at school closure or other NPIs as well? Suggest to rephrase to avoid confusion.
5. Introduce the parameter that are going to be estimated
6. Not very clear of the outcome measure that is used in the study until read the results. Suggest to define in the methods.
7. Line 154-158: strange to define policies here instead of in the introduction.
8. Line 191-196: How did the author define attack rate? Why higher attack rate when children are immune?

·

Basic reporting

This manuscript deals with an emerging topic about assessing the effectiveness of quarantine policies in schools against influenza and COVID-19 by using disease models. The simulation result recommends extending the duration of post-fever isolation to two days rather than the current policy of one day for controlling influenza, and shortening school week down to four days or less to reduce COVID-19 infections. I think these questions to be important, and the tool developed by the authors may be helpful for assessing other intervention policies. To make the proposed method's more visible, I suggest providing the tool information in the manuscript rather than in the supplementary document. Other than that, the following issues should be resolved or at least discussed as its limitations for publication.

Experimental design

1. (Materials and methods section) The authors' compartmental model seems appropriate, but their explanations in the manuscript are too brief. Many modeling details can be founded in the supplementary document. Therefore, I suggest reorganizing the contents in the method section and the supplementary document.

2. (Figure 1) According to model explanations in the supplementary document, heterogeneous contacts between different cohorts were considered in their model. I do not think heterogeneous contacts are necessary for this study, as only school-wide quarantine policies were analyzed. But it seems that including heterogeneous contacts can make their models more general to be reused by other studies. If such a case, it is important to make the proposed method more clear. Figure 1 needs to improve by adding more details of disease model formulations, such as including the cohort index i and redrawing the figure to indicate the mixing between different cohorts.

3. (Lines 170-171) The authors said that ”…reduces the peak prevalence from 20(13-15) to 5 (2-8)…”. However, in figure 2 part A, the medium attack rates are about 22 for the baseline and 7.5 for the two-day policy. The authors should confirm these numbers.

4. Also, the reader may not understand how you measure the outbreak duration. What is the minimum daily infection volume that defines an outbreak? Suggest defining outbreak quantitatively before discussed the simulation results.

5. (Lines 182-183) The medium attack rate is 10% for the COVID-19 baseline. However, in the sensitivity analysis section of the supplementary document (page 2), the authors said that the transmission parameter was adjusted to match up the observed attack rate of 11.3% in a school outbreak, which has a 1.3% difference from the observed data. The authors should explain why there is a discrepancy. Also, it is not clear about parameter tuning details done by the authors. What are the transmission parameters actually used in the simulation experiment?

Validity of the findings

1. (Lines 246-258) The overall conclusion is valid and has connected to research questions and proposed methods. However, the study’s limitations are not presented comprehensively. The follows are important concerns not being well explained in the manuscript:

1. The number of ILIs mainly contain common cold cases caused by respiratory viruses other than influenza viruses. It may be inappropriate to use the number of ILIs to calibrate model parameters directly. Serology testing results or at least any evidence from previous studies would be required.

2. How about the unreported cases? Some cases are infected but not reported. These cases might have a significant impact on epidemiological evaluation and quarantine policies. The authors may propose potential modifications to their model to take such a factor into account or state it as a limitation in their manuscript.

Additional comments

Line 20: Suggest changing the term "SARS-CoV2 " to "SARS-CoV-2".

Table S2: This table only includes a part of the notations used in the proposed model. For example, f¬d is used but not included in this table.

Table 1. Suggest including the baseline policy in this table. Also, the table should include attack rates and outbreak duration rather than only report the decreasing percentages.

Figure 1. Suggest changing the arrow type from double to single for the arrow from the compartment S to compartment V.

Figures 2 and 3. These figures could be better presented by reformatting the background color, x- and y- bars scales, and legends, instead of just copied and pasted from the graphs generated by the online tool.

---

## Round 0.2 · accepted · Accept

All the reviewers' concerns have been correctly addressed, therefore I am pleased to tell you of the acceptance of your paper in its current form in PeerJ. Congratulations!

·

Basic reporting

The authors have made explanations regarding the reviewer questions and commented on related concerns in the resubmitted article. The graphics, tables, and supplementary files have provided sufficient information to enhance the readability of the entire article.

Experimental design

The revised manuscript has clearly stated the simulation model and other materials used in the experimental analysis.

Validity of the findings

The resubmitted manuscript has clearly stated the data, measures, and assumptions, supporting conclusions drawn from the simulation result. The adding statements in the new manuscript are sound and necessary.

Additional comments

No comment.